# Using Machine Learning to Predict Abnormal Carotid Intima-Media Thickness in Type 2 Diabetes

**DOI:** 10.3390/diagnostics13111834

**Published:** 2023-05-23

**Authors:** Chung-Ze Wu, Li-Ying Huang, Fang-Yu Chen, Chun-Heng Kuo, Dong-Feng Yeih

**Affiliations:** 1Division of Endocrinology and Metabolism, Department of Internal Medicine, School of Medicine, College of Medicine, Taipei Medical University, Taipei City 11031, Taiwan; 2Division of Endocrinology and Metabolism, Department of Internal Medicine, Shuang Ho Hospital, Taipei Medical University, New Taipei City 23561, Taiwan; 3Division of Endocrinology and Metabolism, Department of Internal Medicine, Department of Medical Education, Fu Jen Catholic University Hospital, Fu Jen Catholic University, New Taipei City 24352, Taiwan; 4School of Medicine, College of Medicine, Fu Jen Catholic University, New Taipei City 24352, Taiwan; 5Division of Endocrinology and Metabolism, Department of Internal Medicine, Fu Jen Catholic University Hospital, Fu Jen Catholic University, New Taipei City 24352, Taiwan; 6Division of Cardiology, Department of Internal Medicine, Fu Jen Catholic University Hospital, Fu Jen Catholic University, New Taipei City 24352, Taiwan

**Keywords:** machine learning, logistic regression, carotid intima-media thickness, type 2 diabetes mellitus

## Abstract

Carotid intima-media thickness (c-IMT) is a reliable risk factor for cardiovascular disease risk in type 2 diabetes (T2D) patients. The present study aimed to compare the effectiveness of different machine learning methods and traditional multiple logistic regression in predicting c-IMT using baseline features and to establish the most significant risk factors in a T2D cohort. We followed up with 924 patients with T2D for four years, with 75% of the participants used for model development. Machine learning methods, including classification and regression tree, random forest, eXtreme gradient boosting, and Naïve Bayes classifier, were used to predict c-IMT. The results showed that all machine learning methods, except for classification and regression tree, were not inferior to multiple logistic regression in predicting c-IMT in terms of higher area under receiver operation curve. The most significant risk factors for c-IMT were age, sex, creatinine, body mass index, diastolic blood pressure, and duration of diabetes, sequentially. Conclusively, machine learning methods could improve the prediction of c-IMT in T2D patients compared to conventional logistic regression models. This could have crucial implications for the early identification and management of cardiovascular disease in T2D patients.

## 1. Introduction

In recent decades, the prevalence of type 2 diabetes (T2D) has been increasing at a surprising speed, which severely threatens health and life expectancy worldwide [1]. According to statistics from the International Diabetes Federation Diabetes Atlas 2021, there were 536.6 million adults with T2D [2]. This universal trend could also be noted in Taiwan. In 2019, it was estimated that there were two million people with T2D, and this number is increasing by 25,000 patients found every year. Meanwhile, these patients consumed a large portion of the National Health Insurance Policy; the total expense was 1.2 billion USD, approximately 4.66% of the total national budget.

The primary complications of T2D are microvascular and macrovascular diseases. It is estimated that 25–50% of patients with T2D have microvascular diseases [3]. At the same time, it is surprising to note that around 50% of T2D patients die of cardiovascular disease [4]. Thus, detecting, preventing, and treating macrovascular diseases early and accurately are essential tasks for healthcare providers.

Several modalities can evaluate atheroma in arteries, including coronary angiography, stress/rest myocardial perfusion scan, computed tomography angiogram, and treadmill exercise stress test. However, these methods are either expensive or invasive. For example, the treadmill exercise stress test cannot be performed on a large portion of patients who are unable to run due to old age or a variety of causes. In this circumstance, carotid ultrasonography is a non-invasive and inexpensive method that can evaluate intima hyperplasia, atherosclerosis, and stenosis of carotid arteries. Evidence shows that it could be a good predictor of future cardiovascular disease [5].

Based on the information mentioned earlier, it can be agreed that carotid intima-media thickness (c-IMT) is clinically useful. Therefore, it is essential to know the most critical factors affecting c-IMT. Many studies have explored the most important risk factor [6,7]. In an Italian study, systolic blood pressure (SBP) and pulse pressure were associated with changes in c-IMT [8]. However, most of these studies were analyzed using traditional multiple logistic regression (Logit) to assess categorical variables. Artificial intelligence using machine learning (Mach-L), defined as the study of computer algorithms, can improve automatically through experience and by the use of data [9]. It has developed rapidly and has been used in some medical research fields. It enables machines to learn from past data or experiences without being explicitly programmed. It has now become a new modality for data analysis competitive with traditional Logit [10,11,12]. Since Mach-L can capture nonlinear relationships in the data and complex interactions among multiple predictors, it has the potential to outperform Logit for disease prediction [13].

To our knowledge, few studies have tried to predict abnormal c-IMT by Mach-L. Thus, in the present study, we tried to explore two crucial issues in a T2D cohort: 1. Compare the area under the receiver operation curve (ROC) between four different Mach-L methods and traditional Logit. 2. Use Mach-L to predict abnormal c-IMT and rank the importance of risk factors.

## 2. Materials and Methods

### 2.1. Participants and Study Design

The data for this study were derived from the Diabetic Outpatient Clinic at Cardinal Tien Hospital, noted as the “Cardinal Tien Diabetes Study Cohort (CTDSC)”, in Taiwan from 2013 to 2019. We obtained informed consent from all participants and collected data anonymously. The study protocol was approved by the institutional review board of the hospital. In total, 1214 patients with T2D were enrolled. The inclusion criteria were (1) patients aged between 50 and 75 years old, without renal failure on regular hemodialysis, (2) body mass index (BMI) between 22 and 30 kg/m^2^, (3) glycated hemoglobin (HgbA1c) between 6.5 and 10.5%, and (4) all participants had a carotid ultrasound examination before the study. After excluding subjects due to various causes, 924 subjects remained for analysis (495 men and 429 women). The flowchart of participant selection is displayed in Figure 1.

### 2.2. Measurements of Anthropometry and Biochemistry

On the day of the study, a senior nursing staff member recorded the subjects’ medical history, including information about any current medications. Following, a physical examination was performed. Waist circumference (WC) was measured horizontally at the natural waist level. BMI was calculated by dividing the subject’s body weight (kg) by the square of their height (m). SBP and diastolic blood pressure (DBP) were measured using standard mercury sphygmomanometers on the right arm of each subject while they were seated.

Blood samples were taken for biochemical analysis after a 10 h fast. Within one hour of collection, plasma was separated from the blood and stored at 30 °C until analysis of the fasting plasma glucose (FPG) and lipid profiles. The FPG level was measured using a glucose oxidase method (YSI 203 glucose analyzer, Yellow Springs Instruments, Yellow Springs, OH, USA). Total cholesterol and triglyceride (TG) levels were measured using a dry, multilayer, analytical slide method with the Fuji Dri-Chem 3000 analyzer (Fuji Photo Film, Tokyo, Japan). Serum high-density lipoprotein cholesterol (HDL-C) and low-density lipoprotein cholesterol (LDL-C) concentrations were analyzed using an enzymatic cholesterol assay following dextran sulfate precipitation. The urine microalbumin was determined by turbidimetry using the Beckman Coulter AU 5800 biochemical analyzer.

### 2.3. Quantification of Carotid MIT and Plaque

An experienced neurologist blinded to the study measured c-IMT using a high-resolution B-mode ultrasound machine (Philips EPIQ7). He measured c-IMT in the common carotid artery, carotid bulb, and external and internal carotid artery. After reviewing the ultrasound scan video, we determined the most significant extent of plaques visible in the longitudinal view. According to the meta-analysis, we selected the average of the common carotid artery as a representation of the general c-IMT [14]. A c-IMT of less than 1.0 mm is defined as “normal”, while a c-IMT of 1.0 mm or greater is considered “abnormal”.

### 2.4. Description of the Study Data Set

The dependent variables collected from our participants included the following information: age, smoking, alcohol drinking, BMI, duration of diabetes, HgbA1c, HDL-C, LDL-C, ALT, creatinine, SBP, DBP, and microalbuminuria. Smoking and alcohol drinking were categorical variables obtained from the patients’ questionnaires, while all other variables were continuous in nature.

### 2.5. Proposed Scheme

In this study, Table 1 lists the definitions of fifteen baseline clinical variables, including independent variables such as sex, age, BMI, duration of diabetes, smoking, alcohol drinking, HgbA1c, TG, HDL-C, LDL-C, alanine aminotransferase (ALT), creatinine, SBP, DBP, and microalbuminuria. The dependent variable, whether the c-IMT is normal or not, is considered categorical.

In order to assess the applicability of the Mach-L methods in the current study, we selected Logit as the benchmark to compare the AUC values obtained from these methods. It is worth noting that this approach has been utilized in previous studies, as well [15]. Other than the Logit, we proposed a scheme based on four Mach-L methods: classification and regression tree (CART), random forest (RF), Naïve Bayes classifier (NB), and eXtreme gradient boosting (XGBoost). The scheme aims to construct predictive models for predicting normal or abnormal c-IMT and identifying the importance of fifteen risk factors. These Mach-L methods have been widely applied in various healthcare and/or medical informatics applications, and they do not have any prior assumptions about data distribution. Logit is used as a benchmark for comparison.

The first method, CART, is a tree-structured method composed of root nodes, branches, and leaf nodes [16]. Based on the tree structures, it grows recursively from the root nodes and splits at each node using the Gini index to produce branches and leaf nodes according to certain rules. Then, the overgrown tree is pruned to obtain an optimal tree size using the cost–complexity criterion. This operation generates different decision rules that can be used to compose a complete tree structure [17,18].

The second method utilized in this study is RF, which is an ensemble-learning decision trees algorithm that combines bootstrap resampling and bagging [19]. RF’s principle involves randomly generating many different and unpruned CART decision trees, with the decrease in Gini impurity being the splitting criterion, and combining them all to form a forest. The trees in the forest are then averaged or voted on to generate output probabilities and a final model, resulting in the creation of a robust model [20].

The third method used in this study is XGBoost, which is a gradient-boosting technology based on an optimized extension of SGB [21]. Its principle involves sequentially training multiple weak models and ensembling them using the gradient-boosting method of outputs, which achieves better prediction performance. XGBoost uses Taylor binomial expansion to approximate the objective function and arbitrary differentiable loss functions to accelerate the model construction convergence process [22]. Additionally, XGBoost applies regularized boosting techniques to penalize the complexity of the model and correct the overfitting, which increases the model’s accuracy [21].

NB, a popular Mach-L model used for classification tasks, has not been explained yet. This algorithm can sort objects according to specific characteristics and variables based on the Bayes theorem. It calculates the probability of hypotheses on presumed groups [15].

The flowchart of the proposed scheme that combines the four Mach-L methods is demonstrated in Figure 2. In the proposed scheme, we first collected patients’ data to prepare the dataset for model construction. The dataset was randomly split into an 80% training dataset for model building and a 20% testing dataset for out-of-sample testing. In the training process, each Mach-L method has its hyperparameters tuned for constructing a relatively well-performing model. We used a 10-fold cross-validation technique for hyperparameter tuning. To do this, the training dataset was further randomly divided into the training dataset for building the model with a different set of hyperparameters and the validation dataset for model validation. All possible combinations of hyperparameters were investigated by grid search. The model with the lowest root-mean-square error on the validation dataset was viewed as the best model of each Mach-L method. The best models for CART, RF, XGBoost, and NB were generated, and we obtained the corresponding variable importance ranking information.

During the testing process, we used the testing dataset to evaluate the predictive performance of the top-performing models, including CART, RF, XGBoost, and NB. The evaluation of the model’s performance is based on various metrics, such as accuracy, sensitivity, specificity, and area under the curve (AUC), which are presented in Table 2.

To provide a more robust comparison, we randomly repeated the training and testing process mentioned above ten times. The averaged metrics of the CART, RF, XGBoost, and NB models were used to compare the model performances to the benchmark Logit model, which used the same training and testing dataset as the Mach-L methods. A Mach-L model with an average metric lower than that of Logit is considered as a convincing model.

Since all the Mach-L methods used can produce the ranking of each predictor variable in terms of importance, we defined the priorities demonstrated in each model, ranking 1 as the most critical risk factor and 15 as the least critical risk factor. The different Mach-L methods may produce different variable importance rankings due to their unique modeling characteristics. To enhance the stability and integrity of re-ranking the importance of risk factors, we integrated the variable importance ranking of the convincing Mach-L models. In the final stage of the proposed scheme, we summarized and discussed our significant findings about convincing Mach-L models and identified critical variables.

In this study, we carried out all methods using R software version 4.0.5 and RStudio version 1.1.453 with the required packages installed. The implementations of RF, CART, XGBoost, and NB are “randomForest” R package version 4.6–14, “gbm” R package version 2.1.8, “rpart” R package version 4.1–15, and “XGBoost” R package version 1.5.0.2, respectively. Additionally, we used the “caret” R package version 6.0–90 to estimate the best hyperparameter set for the developed effective CART, RF, XGBoost, and NB methods. The “stats” R package version 4.0.5 was used to implement Mach-L with the default setting used to construct the models.

## 3. Results

A total of 924 patients with T2D (495 men and 429 women) were enrolled. Their baseline characteristics are shown in Table 1. When comparing the risk factors between different genders, we found that women were older than men. Additionally, the HDL-C level was significantly lower in men than in women, while serum creatinine, ALT, and DBP were significantly higher in men than in women. Table 2 shows the baseline characteristics of subjects with normal or abnormal c-IMT. Comparing those with normal c-IMT to those with abnormal c-IMT, the latter group had significantly older age, lower BMI, and higher creatinine and DBP.

The results of the comparison between traditional Logit and four Mach-L methods, including RF, CART, XGBoost, and NB, in predicting abnormal c-IMT are shown in Table 3. Based on the AUC of all methods in Table 3, the predictability of the Mach-L methods was similar to that of traditional Logit, except for CART. Therefore, it can be concluded that apart from Logit, the other three Mach-L methods can also be utilized in medical research and for selecting risk factors, allowing for comparisons with other well-established studies.

Table 4 illustrates the average rank value of each risk factor generated by the Logit, RF, CART, XGBoost, and NB methods. The table shows that different Mach-L methods generated different relative importance rank values for each risk factor. To fully integrate the importance ranks of each risk factor in all Mach-L and Logit methods, the average importance value of each risk factor was obtained by averaging the ranking values of each variable in each method. It should be noted that the darkness of the color indicates the importance of the risk factors. The darker the color, the higher the importance of the risk factor. For instance, in the RF method, the three most important factors are baseline age, sex, and baseline creatinine.

Figure 3 depicts the risk factors based on the increasing order of their average rank values. It can be noted from the figure that the six most important risk factors in predicting abnormal c-IMT in a 4-year-follow-up T2D cohort were baseline age, sex, creatinine, BMI, DBP, and duration of diabetes.

## 4. Discussion

The present study had two aims. The first was to compare the accuracy of Logit and different Mach-L methods. The second was to identify the importance rank of risk factors for abnormal c-IMT. Our data showed that the Mach-L methods outperformed the traditional Logit method, particularly in the accuracy and specificity of prediction. Additionally, we identified sex, age, duration of diabetes, BMI, creatinine, and DBP as the most critical factors.

It is estimated that 25–50% of patients with T2D have microvascular disease. The underlying pathophysiology has been studied in other research. Individuals diagnosed with T2D are at a significantly higher risk of developing cardiovascular disease. In fact, atherosclerosis is responsible for more deaths among diabetic patients than any other causes combined. Some patients with multiple risk factors tend to cluster in a syndrome referred to as metabolic syndrome. The advanced glycation end-product is one mechanism that links hyperglycemia and atherogenesis. Hyperglycemia increases the linkage of glucose to proteins, resulting in insoluble complexes known as advanced glycation end-products, which cause changes in endothelial cells. Elevated TG levels in diabetic patients are also risk factors for cardiovascular disease. Although LDL-C levels may not necessarily be high in T2D, higher levels (or LDL phenotype B) are more atherogenic. The relationship between obesity and hypertension is well-documented, and obesity can exacerbate other risk factors [23].

The results of our current study are consistent with those of Peczyńska J et al., who discovered that increased c-IMT was linked to higher BMI, blood pressure, HbA1c, lipids, hsCPR, and NT-proBNP [24]. It is important to note that they concluded their study on type 1 diabetes patients. In Taiwan, the incidence of type 1 diabetes is only approximately 1% [25]. Therefore, we excluded type 1 diabetes in our study. Hypothetically, the underlying causes of type 1 diabetes are distinct from those of T2D and, as a result, should be discussed separately. Nonetheless, comparing our study to the Peczyńska study, it can be observed that there are several common risk factors. Additionally, in their study, they discovered other new markers related to c-IMT, which is quite intriguing and promising. Unfortunately, we did not collect comparable data in our current study.

Type 1 diabetes is characterized by absolute insulin deficiency, while T2D involves relative insulin deficiency and insulin resistance. Despite both diseases sharing hyperglycemia as a clinical presentation, they are fundamentally different. Therefore, we hypothesize that while type 1 diabetes and T2D may share some risk factors for abnormal c-IMT, there must also be additional risk factors that are not the same. With a sufficient number of patients with type 1 diabetes, it would be possible to use the same Mach-L methods to explore this interesting question. Various Mach-L methods are currently employed in medical research, and in our study, we selected the most commonly used and well-documented four methods.

Traditionally, the Logit method is widely used to analyze continuous variables in medical research. However, Logit is difficult to use when assessing nonlinear data patterns, and effectively using the Logit method requires fitting its strong model assumptions during modeling. In contrast to Logit, the Mach-L methods do not need strong model assumptions and can capture delicate underlying nonlinear relationships contained in empirical data [26]. Based on our present data, the Mach-L methods are superior to traditional Logit in terms of the accuracy and specificity of the RF, XGBoost, and NB methods. Our results suggest that Mach-L methods might provide a new predictive tool in medical research.

The most critical risk factor in the present study was age. This result was in concordance with the majority of previous studies [27,28]. In a meta-analysis, van den Munckhof et al. demonstrated a gradual and linear increase in the relationship between age and c-IMT (r = 0.91, *p* < 0.001) in 10,124 subjects from 58 studies [27]. It is not surprising that the subjects in their study were healthy, without significant cardiovascular risk factors. On the other hand, the participants in our cohort were patients with T2D who had multiple risk factors. Glycemic control, lipid profile, and blood pressure in T2D are well-known to contribute to an increase in c-IMT. The age-related abnormal c-IMT may be attributed to the progressive increase in LDL-C. The causation is supported by the Multi-Ethnic Study of Atherosclerosis, which enrolled community-dwelling adults aged 45–84 years old in the USA [29]. They found that each standard deviation (SD) increase in LDL-C would increase c-IMT by 0.037 mm (95% confidence interval: 0.018–0.055) in 45–54 years old participants. The association increased with age and became 0.087 mm per SD in LDL-C in the group of 75–84-year-olds.

The second most important factor contributing to c-IMT in our results was sex. Boulos et al. investigated 9347 women and 12,676 men and found that for subjects aged <45 to ≥80 years, the prevalence of abnormal c-IMT ranged from 0.13% to 29.3% in women and 0.6% to 40.1% in men [28]. According to their results, the prevalence of abnormal c-IMT increased linearly with age. Additionally, the prevalence of abnormal c-IMT in men was higher than in women across all age intervals. The gap in prevalence between different genders increased with age and reached its largest value in the 75–79 age range. Although our findings were not entirely novel, there is still a controversial conclusion to this dilemma. This phenomenon is not difficult to explain. Abd El-Hafez H. et al. conducted a study that showed that abnormal c-IMT was independently associated with men [30]. They hypothesized that higher LDL-C and WC and lower HDL-C were found in men. They also pointed out that this difference between men and women became less significant after menopause in women. However, it should be noted that their study was quite small (men = 37, women = 38). Additionally, all participants in their investigation were obese. This study indicates that traditional risk factors, such as components of metabolic syndrome, might contribute to the gender difference of c-IMT. As for the special group of T2D, the results of Ogbera were of contribution [31]. He indicated that the prevalence of metabolic syndrome was similar in men and women (83% and 86%, respectively) in a cohort of 963 patients with T2D. To further explore whether there are different abnormal components in this particular group, the research of Yang et al. might be of importance [32]. Since their study was conducted in Korea, which has a similar Asian ethnic group as China, the results might be more convincing. In a cohort of 11,136 subjects, they found that men had higher risks of hypertension, glucose dysregulation, and hyper-TG. On the other hand, women had a higher chance of having low HDL-C than men. The higher number of metabolic-syndrome components might explain the results of the present study.

The third most important risk factor was the creatinine level, which was positively related to c-IMT. Our finding was compatible with previous studies [33,34]. Gentile et al. demonstrated that an abnormal creatinine level (>1 mg/dL) is associated with a higher c-IMT (≥1.2 mm), with an odds ratio of 4.12 (95% confidence interval: 1.22–13.86) [35]. This is a substantial correlation. However, their study’s limitation was that only 310 women participated, which might be less persuasive. This relationship could be explained by the fact that abnormal renal function in T2D is also associated with atherogenic factors such as higher TG, lower HDL-C, and increased apolipoprotein AI [36]. Our study is the only one that was conducted in patients with T2D and is of clinical value.

The fourth most important factor in line is BMI. The relationship between obesity and c-IMT has been demonstrated in many previous studies [37,38,39]. In the Atherosclerosis Risk in Community Study, 13,282 subjects were enrolled and followed-up on for two years. The body weight increased from 9.7 to 20.8 kg in different genders and ethnic groups (black and white). The results showed that for every 10 kg increment, there was a 0.002 to 0.016 mm increment of c-IMT. The authors explained that the increase of c-IMT with BMI was due to several risk factors, such as hyperglycemia, hyperinsulinemia, hypertension, and reduced HDL-C, all of which are related to body weight [39]. However, their cohort enrolled not only patients with T2D but also subjects without T2D, so extrapolating their results to the present study should be exercised with caution. Nevertheless, our study proved the crucial association between BMI and c-IMT in the T2D cohort. Future studies with a longitudinal design are needed to further confirm this relationship.

The fifth most important factor was DBP. A large amount of research has shown that blood pressure is associated with c-IMT, especially in patients with T2D [40,41]. Itoh et al. evaluated the effects of both SBP and DBP in a Japanese cohort of 1241 individuals [42]. The abnormal carotid plaque (c-IMT ≥ 1.1 mm) increased as blood pressure became higher, even after adjusting for other covariates. Similarly, another Chinese study with 3789 subjects aged ≥45 years without cardiovascular disease also helped confirm our data [43]. They found that SBP and DBP were positively related to c-IMT. It is interesting to note that the increase in c-IMT is reversible in patients with T2D. By taking antihypertensive agents and controlling hypertension well, the increase of c-IMT could be inhibited, which was independent of glycemic control [44]. In the present study, we found that DBP is an important factor of c-IMT. SBP and DBP may have different effects on c-IMT. DBP reflects the pressure that the blood exerts on the walls of the arteries when the heart is at rest, and this pressure can lead to changes in the structure of the arterial wall over time.

The last important factor to consider is the duration of T2D. In a previous study, the Insulin Resistance Atherosclerosis Study (IRAS) enrolled 489 patients with T2D and found no relationship between the duration of T2D and c-IMT [45]. Instead, FPG was identified as the only contributor to c-IMT. It should be noted that this study was conducted before 1997, when glycated hemoglobin was not widely used as a more reliable measurement of glycemic control. Therefore, their findings are not compatible with the general recognition that the duration of T2D is an important factor in T2D-related complications. For example, the duration influence has been repeatedly shown in many studies about type 1 diabetes [46,47,48]. Interestingly, few studies have evaluated this topic in T2D. The only one, to our knowledge, was conducted by Zoungas et al. [49]. They showed that in 11,140 T2D patients, both macrovascular and microvascular diseases were associated with the duration of T2D. The average duration of T2D was 7.9 years. This finding is persuasive since their sample size was large and could support our present study.

There were some limitations in our study. First, the use of c-IMT as a primary outcome parameter has inherent limitations. This is due to the presence of both inter- and intra-observer bias when measuring c-IMT [24]. In addition, the comprehensive data analysis and interpretation of results may differ based on the methodology used. Second, many risk factors contribute to c-IMT but were not included. For example, a family history of cardiovascular disease is an important factor for predicting c-IMT. Subjects with a family history of cardiovascular disease have more c-IMT than those without a family history [50]. Moreover, high-sensitivity C-reactive proteins and proinflammatory cytokines have also been associated with c-IMT in several studies. Third, patients with T2D receiving treatment of sodium–glucose cotransporter 2 inhibitor or glucagon-like peptide-1 receptor agonist may have reduced cardiovascular mortality in clinical trials. Our data lack drug information, which might also contribute to the c-IMT. Although we acknowledge the limitations of our data, we believe that the present study demonstrates that Mach-L outperforms Logit in predicting T2D c-IMT, and the six most important risk factors are identified as aforementioned.

Currently, c-IMT is not a standard method for assessing cardiovascular disease in patients with T2D. However, as previously mentioned, it is a less expensive, non-invasive, and less labor-intensive modality. There have been several studies published in this field, such as the study conducted by Polak et al. [51]. In this study, Polak et al. demonstrated that c-IMT could predict future cardiovascular disease in 2965 subjects of the Framingham Offspring Study after a follow-up period of 7.2 years. Therefore, we believe that c-IMT is still a valuable surrogate measure for evaluating the risk of cardiovascular disease.

## 5. Conclusions

In conclusion, Mach-L methods exhibited higher accuracy and specificity compared to traditional Logit. Sex, age, duration of diabetes, BMI, creatinine, and DBP were identified as the most significant risk factors in Chinese patients with T2D for predicting c-IMT.

## Figures and Tables

**Figure 1 diagnostics-13-01834-f001:**
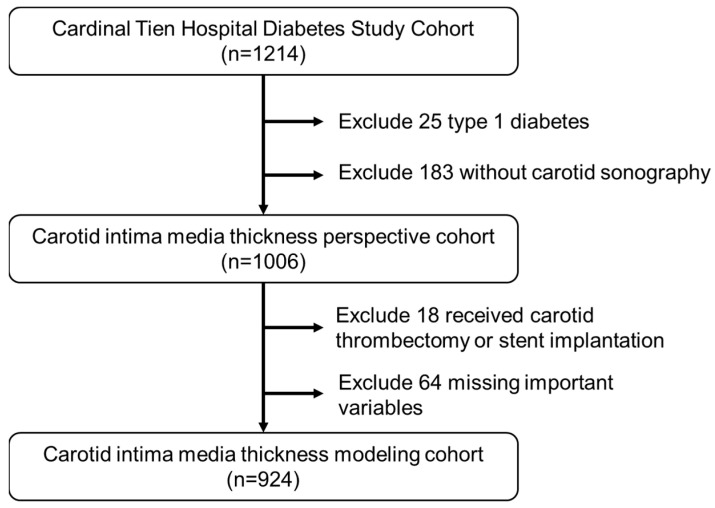
Flowchart of sample selection from the Cardinal Tien Hospital Diabetes Study Cohort.

**Figure 2 diagnostics-13-01834-f002:**
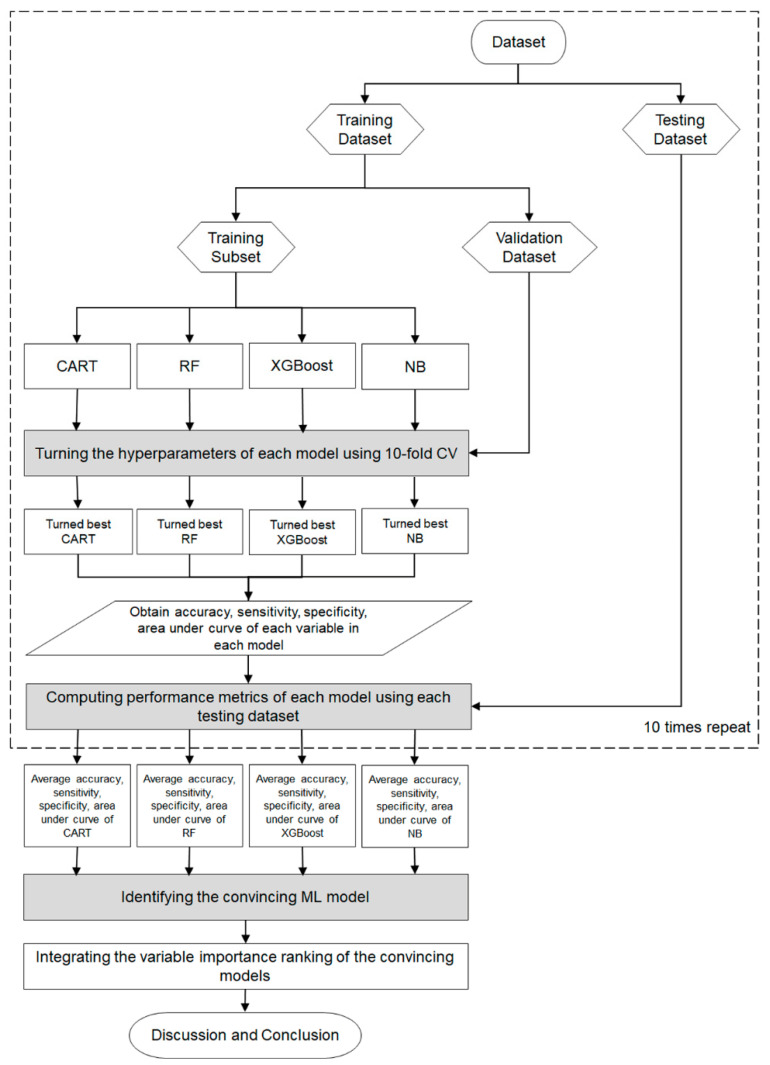
Proposed machine learning scheme in the cohort.

**Figure 3 diagnostics-13-01834-f003:**
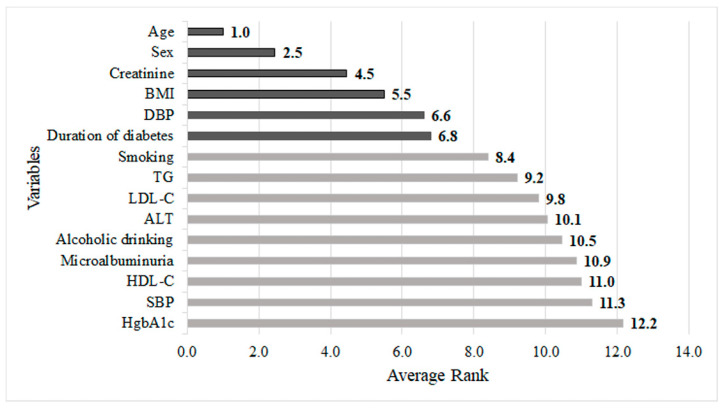
Integrated importance ranking of all risk factors predicting carotid intima-media thickness in patient with type 2 diabetes.

**Table 1 diagnostics-13-01834-t001:** Baseline characteristics of subjects in different genders.

	Men	Women
*n*	495	429
Age (y) ***	63.8 ± 10.9	66.5 ± 10.0
Smoking (*n* (%))	191 (38.5%)	15 (3.5%)
Alcoholic drinking (*n* (%))	53 (10.7%)	1 (0.2%)
BMI (Kg/m^2^)	26.3 ± 3.6	26.6 ± 4.4
Duration of diabetes (y)	15.3 ± 7.5	15.6 ± 7.8
HgbA1c (%)	7.5 ± 1.5	7.5 ± 1.3
TG (mg/dL)	140.0 ± 120.1	133.0 ± 72.6
HDL-C (mg/dL) ***	41.7 ± 11.1	47.4 ± 10.7
LDL-C (mg/dL)	96.6 ± 26.6	97.5 ± 28.3
ALT (U/dL) ***	31.5 ± 22.1	25.6 ± 16.5
Creatinine (mg/dL) *	1.05 ± 0.48	0.76 ± 0.55
SBP (mmHg)	130.0 ± 13.3	130.9 ± 14.6
DBP (mmHg) *	76.5 ± 8.2	74.5 ± 15.1
Microalbuminuria (mg/g)	180.4 ± 689.2	114.0 ± 606.5

Data showed as mean ± SD, BMI: body mass index, HgbA1c: glycated hemoglobin, TG: triglyceride, HDL-C: high-density lipoprotein-cholesterol, LDL: low-density lipoprotein-cholesterol, ALT: alanine aminotransferase, SBP: systolic blood pressure, DBP: diastolic blood pressure. * *p* < 0.05, *** *p* < 0.001.

**Table 2 diagnostics-13-01834-t002:** Baseline characteristics of subjects between normal and abnormal carotid intima-media thickness.

	Normal	Abnormal
*n*	710	214
Age (y) ***	63.4 ± 10.3	70.6 ± 9.5
Smoking (*n* (%))	150 (21.1%)	56 (26.2%)
Alcoholic drinking (*n* (%))	47 (6.6%)	7 (3.3%)
BMI (Kg/m^2^) **	26.6 ± 4.0	25.8 ± 3.6
Duration of diabetes (y)	14.8 ± 7.1	17.4 ± 9.0
HgbA1c (%)	7.5 ± 1.4	7.5 ± 1.4
TG (mg/dL)	138.5 ± 108.1	131.1 ± 74.9
HDL-C (mg/dL)	44.6 ± 10.9	43.4 ± 12.3
LDL-C (mg/dL)	97.2 ± 27.9	96.2 ± 25.8
ALT (U/dL)	25.2 ± 19.7	27.4 ± 20.7
Creatinine (mg/dL) ***	0.87 ± 0.51	1.05 ± 0.59
SBP (mmHg)	130.0 ± 14.0	132.0 ± 13.4
DBP (mmHg) *	76.2 ± 12.8	73.7 ± 8.2
Microalbuminuria (mg/g)	129.8 ± 613.5	216.9 ± 767.7

Data showed as mean ± SD. BMI: body mass index, HgbA1c: glycated hemoglobin, TG: triglyceride, HDL-C: high-density lipoprotein-cholesterol, LDL: low-density lipoprotein-cholesterol, ALT: alanine aminotransferase, SBP: systolic blood pressure, DBP: diastolic blood pressure. * *p* < 0.05, ** *p* < 0.01, *** *p* < 0.001.

**Table 3 diagnostics-13-01834-t003:** Comparison of accuracy, sensitivity, specificity, and AUC among multiple logistic regression (Logit) and machinal learning methods by receiver operating characteristic curve.

	Accuracy	Sensitivity	Specificity	AUC
Logit	0.669 ± 0.081	0.682 ± 0.116	0.665 ± 0.134	0.692 ± 0.030
CART	0.523 ± 0.082	0.488 ± 0.040	0.532 ± 0.112	0.511 ± 0.036
RF	0.703 ± 0.071	0.622 ± 0.122	0.724 ± 0.132	0.692 ± 0.036
XGBoost	0.716 ± 0.048	0.616 ± 0.123	0.742 ± 0.097	0.688 ± 0.030
NB	0.683 ± 0.059	0.664 ± 0.094	0.688 ± 0.101	0.692 ± 0.029

Data showed as mean ± SD; AUC: area under curve, CART: classification and regression tree, RF: random forest, XGBoost: eXtreme gradient boosting, NB: Naïve Bayes classifier.

**Table 4 diagnostics-13-01834-t004:** Importance rank value of each risk factor using the four machine learning methods.

	Logit		CART	RF	XGBoost	NB
Sex	2.0		2.0	2.4	1.0	2.5
Age	1.0		6.0	1.0	2.0	1.0
BMI	7.8		9.0	7.1	3.0	3.3
Duration of diabetes	12.3		5.0	7.0	4.0	7.1
Smoking	7.4		3.0	5.4	5.0	11.4
Alcoholic drinking	10.2		1.0	9.8	6.0	10.5
HgbA1c	10.2		7.0	11.4	7.0	13.2
TG	8.1		4.0	11.7	8.0	7.4
HDL-C	11.9		8.0	11.0	9.0	11.1
LDL-C	13.1		15.0	10.2	10.0	9.2
ALT	5.4		15.0	8.1	11.4	11.5
Creatinine	10.4		15.0	3.5	12.3	5.4
Microalbuminuria	11.4		15.0	12.1	13.2	10.3
SBP	5.7		15.0	9.6	14.4	12.4
DBP	3.1		15.0	9.7	15	3.7
Rank value	1.0~1.4	1.5~2.4	2.5~3.4	3.5~4.4	4.5~5.4	5.5~

Abbreviations as the footnotes in Table 1 and Table 3.

## Data Availability

Data are available upon request due to privacy/ethical restrictions.

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
