# Peer review of "Using Machine Learning to Predict Abnormal Carotid Intima-Media Thickness in Type 2 Diabetes"

_diagnostics, 2023, doi:10.3390/diagnostics13111834_

Round 1
Reviewer 1 Report
First of all, the accuracy, sensitivity, specificity, and AUC are all very low. I don't think 70% accuracy on diagnosis would be meaningful in the medical field.
Second, what is presented is nothing but correlation. Correlation doesn't mean causation. What is the deep medical reason for the correlation? The discussion should be strengthened with medical analysis as the foundation for the correlation.
Third, the dataset description is not clear.
Author Response
We would like to express our gratitude to the reviewer for taking the time to review our article and providing valuable feedback. We have carefully considered the reviewer's recommendations and opinions. We appreciate the reviewer's efforts to help us improve the quality of our article. We would like to take the reviewer's questions and comments into account and have made the necessary corrections and revisions to the article. We have answered the reviewer's questions point to point and made sure that our article is now more accurate and informative.
- First of all, the accuracy, sensitivity, specificity, and AUC are all very low. I don't think 70% accuracy on diagnosis would be meaningful in the medical field.
Response: Thank you for your valuable comment. As mentioned in the introduction, our study had two goals: 1) to compare the accuracy of traditional logistic regression and four different Mach-L methods, and 2) to rank the importance of risk factors and compare them to other studies. We found that, except for CART, the other three Mach-L methods were non-inferior to logistic regression. This finding led us to conclude that Mach-L might be suitable for this type of medical research as it can capture non-linear relationships between variables. The second goal was also achieved by ranking the risk factors and providing clear descriptions in the discussion. Our study has several strengths over similar ones; we had a large sample size, used Mach-L methods, and focused on patients with type 2 diabetes. To our knowledge, few studies have done so with similar novelty. While we agree it is only a cross-sectional study, we plan to conduct a longitudinal one in the future. However, we also want to emphasize that the present study was conducted in Chinese, and caution must be exercised when extrapolating to other ethnic groups.
The reviewer has raised a concern regarding the AUC of around 0.7, stating that it is not convincing. We concur that the sensitivity and specificity pertaining to this AUC are unsatisfactory. Nevertheless, in medical research, an AUC in this range may still be considered worthy of publication. In support of our viewpoint, we present two examples (Lewandowska KB, et al. Diagnostics 2023;13(6):1109; Yu Y, et al. BMC Anesthesiol. 2022;22(1):22), dealing with different topics and areas, but having similar AUC values of around 0.7. As highlighted in the previous paragraph, our aim was to validate the use of Mach-L in medical research and compare its risks with those of traditional statistical methods. It is noteworthy that our data exhibit some unique features and could therefore be deemed suitable for publication.
- Second, what is presented is nothing but correlation. Correlation doesn't mean causation. What is the deep medical reason for the correlation? The discussion should be strengthened with medical analysis as the foundation for the correlation.
Response: We appreciate the reviewer's comments regarding our cross-sectional study. While we agree that our results do not establish causality, we have re-evaluated the relationships between risk factors and c-IMT and found sex, age, duration of diabetes, BMI, creatinine, and DBP to be the most significant factors. Although these risk factors have been demonstrated in previous studies, the novelty of our data lies in the ranking of their importance. For instance, our data suggest that gender has the most profound effect, followed by age, which surpasses the importance of diabetes, including both duration and blood glucose level. Interestingly, glucose control was not selected by Mach-L, meaning that abnormal c-IMT in diabetic patients is primarily determined by gender, age, and duration of diabetes rather than glucose control. In addition, SBP was not among the top six risk factors, which is surprising. However, we believe that in diabetic patients, the influence of SBP is mainly diluted by DBP. These findings have not been previously reported and are therefore valuable.
The underlying causes of these relationships are discussed at the end of each factor's discussion. For instance, our analysis indicates that age-related abnormal c-IMT may be attributed to the progressive increase in LDL-C. This causation is supported by the Multi-Ethnic Study of Atherosclerosis, which enrolled community-dwelling adults aged 45-84 years in the USA (Polak JF, et al. J Am Heart Assoc. 2013;2(2):e000087. PMID: 23568342). They found that per standard deviation (SD) increase in LDL-C, c-IMT increased by 0.037 mm (95% confidence interval: 0.018-0.055) in participants aged 45-54 years, and the association became 0.087 mm per SD in the 75-84 age group. The reviewer may find similar descriptions in each of the paragraphs.
- Third, the dataset description is not clear.
Response: We deeply appreciate the reviewer's comments, and we have carefully re-examined our study cohort's description. As suggested by the reviewer, we have included the following paragraph to reinforce the dataset's description in the Materials and Methods section, 2.4. Description of the study data set on Page 5 Line 13-18
Once again, we would like to thank the reviewer for their time and effort in reviewing our article. Their comments and suggestions have been immensely helpful in improving the quality of our work. We hope that our revised article meets the high standards of your esteemed journal.
Sincerely,
Dong-Feng Yeih
Division of Cardiology, Department of Internal Medicine, Fu Jen Catholic University Hospital; School of Medicine, College of Medicine, Fu Jen Catholic University, New Taipei City 24352, Taiwan, R.O.C.
Tel: 886-2-8512-8888 ext 28751
E-mail: ocean082977@gmail.com
Reviewer 2 Report
1. Logistic regression has become an important aspect of machine learning. The author compared machine learning with logistic regression. The article design is incorrect.
2. The evaluation indicators for evaluating the effectiveness of the model should try to use AUC instead of accuracy. This article shows that other algorithms have no advantages compared to logistic regression.
3. Compared to non-invasive evaluation methods, this study has no clinical application value.
No comments
Author Response
We would like to express our gratitude to the reviewer for taking the time to review our article and providing valuable feedback. We have carefully considered the reviewer's recommendations and opinions. We appreciate the reviewer's efforts to help us improve the quality of our article. We would like to take the reviewer's questions and comments into account and have made the necessary corrections and revisions to the article. We have answered the reviewer's questions point to point and made sure that our article is now more accurate and informative.
- Logistic regression has become an important aspect of machine learning. The author compared machine learning with logistic regression. The article design is incorrect.
Response: We respect the reviewer’s comment very much. However, as we have mentioned in our method:
‘However, most of these studies were analyzed using traditional multiple logistic regression (Logit) to assess categorical variables. Artificial intelligence using machine learning (Mach-L), defined as the study of computer algorithms, can improve automatically through experience and by the use of data [9]. It has developed rapidly and has been used in some medical research fields. It enables machines to learn from past data or experiences without being explicitly programmed. It has now become a new modality for data analysis competitive with traditional Logit [10-12]. Since Mach-L can capture nonlinear relationships in the data and complex interactions among multiple predictors, it has the potential to outperform Logit for disease prediction [13].’
In this paragraph, the most important concepts that Mach-L differs from the Logistic regression are: 1. It does not need hypotheses for data such as normal distribution 2. Mach-L could capture non-linear relationship better compare to logistic prediction. We also gave a reference [13] in the manuscript.
Upon examining the history of logistic regression and Mach-L, it is evident that there is a 30-year gap between the development of logistic regression and the first instance of Mach-L employed in this study. Logistic regression was originally proposed by Berkson in 1944. Conversely, classification and regression tree was developed in 1972 as part of the THAID project by Messenger and Mandell. Random forest was introduced in 1995, eXtreme gradient boosting in 2016, and Naïve Bayes in 2011.
- The evaluation indicators for evaluating the effectiveness of the model should try to use AUC instead of accuracy. This article shows that other algorithms have no advantages compared to logistic regression.
Response: We agree with the reviewer’s comment. In the Result Section, we mentioned that three out of the four Mach-L approaches exhibited non-inferior AUC compared to the conventional logistic regression, while providing better accuracy. As the AUC for logistic regression and Mach-L were comparable, we selected accuracy as the metric to showcase the efficacy of Mach-L. However, we do acknowledge the significance of AUC.
- Compared to non-invasive evaluation methods, this study has no clinical application value.
Response: Thank you for your valuable comment. As stated in the introduction, our study had two objectives: firstly, to compare the accuracy of traditional logistic regression and four different machine learning (Mach-L) methods (as explained in the first response), and secondly, to rank the significance of risk factors and compare them with other studies. Our findings revealed that, apart from CART, the other three Mach-L methods were comparable to logistic regression. This conclusion implies that Mach-L could be appropriate for medical research of this nature as it can capture non-linear relationships between variables. We also achieved our second goal by ranking the risk factors and offering clear explanations in the discussion. Our study has several strengths over similar studies: we had a large sample size of Chinese patients, used Mach-L methods, and focused solely on patients with type 2 diabetes. To the best of our knowledge, few studies have explored this with similar originality. Our data indicates that there is a gender difference in the formation of atheroma in the carotid artery, and age is the second most significant factor. This implies that age is more important than diabetes itself. It is also noteworthy that glucose control (as measured by glycated hemoglobin level) is not as significant as the duration of diabetes. We believe that all of this information is valuable for clinical use. We request you to consider our statement of the strength of this article and kindly accept this manuscript for publication, if possible.
Once again, we would like to thank the reviewer for their time and effort in reviewing our article. Their comments and suggestions have been immensely helpful in improving the quality of our work. We hope that our revised article meets the high standards of your esteemed journal.
Sincerely,
Dong-Feng Yeih
Division of Cardiology, Department of Internal Medicine, Fu Jen Catholic University Hospital; School of Medicine, College of Medicine, Fu Jen Catholic University, New Taipei City 24352, Taiwan, R.O.C.
Tel: 886-2-8512-8888 ext 28751
E-mail: ocean082977@gmail.com
Reviewer 3 Report
This is a nicely written original paper on the use machine learning to predict abnormal carotid intima-media thickness in patients with type 2 diabetes.The paper is well presented. It covers an important issue of programming, artificial intelligence and machine learning in order to facilitate work of clinical practicioners. In addition, the developement of vascular complications, like carotid intima-media thickening (CIMT) and plaques, is often present in young adults with diabetes. The co-existance of CIMT and plaques in young diabetic patients expose them at higher risk of thrombotic complications like ischemic stroke, and myocardial infarction, despite young age.
I believe that this paper could be considered for publications, however I would like the Authors to respond to a few comments.
Suggestions:
1. I would advise to change in the title term " medium intima thickness of carotid artery" to 'carotid intima-media thickness'. The term carotid intima-media is easy to identify for the researchers working in this field, and could attract more attention, thus gaining citations.
2. The Authors stated that: "It is estimated that 25-50% of patients with T2D have microvascular diseases". I do agree that this is important worldwide burden. Please address this clinical problem, more in depth, in discussion.
3. I believe that it is worth of noting that also adolescents and young adults with T1D have a high risk of having micro and macrovascular complications despite young age. The Authors may find utile to address this issue published by Peczyńska J et al. (J. Clin. Med. 2022, 11(16), 4732; https://doi.org/10.3390/jcm11164732). In this original paper, increased CIMT values were associated with a significantly higher concentration of HbA1c, lipid levels, hsCRP and NT-proBNP, BMI and blood pressure values. The Authors concluded that, youth with T1D and vascular complications present with many abnormalities in the classical and new CVD biomarkers. hsCRP and MPO seem to be the most important markers for estimating the risk of macroangiopathy. NT-proBNP may present a possible marker of early myocardial injury in this population.
Please, address this issue. Do the Authors of the present paper think that their algorithm could be used also in T1D, or a different machine learning methods should be used.
4. Please, add paragraph on the CIMT limitations. This issue was described in depth in J. Clin. Med. 2021, 10(20), 4628; https://doi.org/10.3390/jcm10204628
Anyway, a routine CIMT measurements is not a standard for the assessment of cardiovascular risk according to current statements of the respective Societies. Please, write why You think that it is worth of measuring.
Author Response
We would like to express our gratitude to the reviewer for taking the time to review our article and providing valuable feedback. We have carefully considered the reviewer's recommendations and opinions. We appreciate the reviewer's efforts to help us improve the quality of our article. We would like to take the reviewer's questions and comments into account and have made the necessary corrections and revisions to the article. We have answered the reviewer's questions point to point and made sure that our article is now more accurate and informative.
- I would advise to change in the title term " medium intima thickness of carotid artery" to 'carotid intima-media thickness'. The term carotid intima-media is easy to identify for the researchers working in this field, and could attract more attention, thus gaining citations.
Response: We appreciate the reviewer's comment. We corrected our title to “Using machine learning to predict abnormal carotid intima-media thickness in type 2 diabetes”.
- The Authors stated that: "It is estimated that 25-50% of patients with T2D have microvascular diseases". I do agree that this is important worldwide burden. Please address this clinical problem, more in depth, in discussion.
Response: Thank you for your great comment and we have added the following paragraph to respond to your comment on Page 9 Line 7-19.
“It is estimated that 25-50% of patients with T2D have microvascular disease. The underlying pathophysiology has been studied in other research. Individuals diagnosed with T2D are at a significantly higher risk of developing cardiovascular disease. In fact, atherosclerosis is responsible for more deaths among diabetic patients than any other cause combined. Some patients have multiple risk factors that tend to cluster in a syndrome referred to as a metabolic syndrome. Advanced glycation end-product is one mechanism that links hyperglycemia and atherogenesis. Hyperglycemia increases the linkage of glucose to proteins, resulting in insoluble complexes known as advanced glycation end-products, which cause changes in endothelial cells. Elevated TG levels in diabetic patients are also risk factors for cardiovascular disease. Although LDL-C levels may not necessarily be high in T2D, higher levels (or LDL phenotype B) have been shown to be more atherogenic. The relationship between obesity and hypertension is well-documented, and obesity can exacerbate other risk factors”
- I believe that it is worth of noting that also adolescents and young adults with T1D have a high risk of having micro and macrovascular complications despite young age. The Authors may find utile to address this issue published by Peczyńska J et al. (J. Clin. Med. 2022, 11(16), 4732; https://doi.org/10.3390/jcm11164732). In this original paper, increased CIMT values were associated with a significantly higher concentration of HbA1c, lipid levels, hsCRP and NT-proBNP, BMI and blood pressure values. The Authors concluded that, youth with T1D and vascular complications present with many abnormalities in the classical and new CVD biomarkers. hsCRP and MPO seem to be the most important markers for estimating the risk of macroangiopathy. NT-proBNP may present a possible marker of early myocardial injury in this population.
Response: Thank you for your great comment and we have added the following paragraph to respond to your comment on Page 9 Line 20-29.
“The results of our current study are consistent with those of PeczyÅ„ska J et al., who discovered that increased c-IMT was linked to higher BMI, blood pressure, HbA1c, lipids, hsCPR, and NT-proBNP. It is important to note that their study was conducted on Type 1 diabetes patients. In Taiwan, the incidence of Type 1 diabetes is only approximately 1%. This is why we excluded Type 1 diabetes in our study. Hypothetically, the underlying causes of Type 1 diabetes are distinct from those of T2D and, as a result, should be discussed separately. Nonetheless, from the PeczyÅ„ska study, it can be observed that there are several common risk factors compared to our study. Additionally, in their study, they discovered other new markers related to c-IMT, which is quite intriguing and promising. Unfortunately, we did not collect comparable data in our current study.”
- Please, address this issue. Do the Authors of the present paper think that their algorithm could be used also in T1D, or a different machine learning methods should be used.
Response: Thank you for your excellent comment. we have added the following paragraph to respond to your comment on Page 9 Line 30-31 and Page 10 Line 1-6.
“Type 1 diabetes is characterized by absolute insulin deficiency, while T2D involves relative insulin deficiency and insulin resistance. Despite both diseases sharing hyperglycemia as a clinical presentation, they are fundamentally different. Therefore, we hypothesize that while type 1 diabetes and T2D may share some risk factors for abnormal c-IMT, there must also be additional risk factors that are not the same. With a sufficient number of patients with type 1 daibetes, it would be possible to use the same Mach-L methods to explore this interesting question. Various Mach-L methods are currently employed in medical research, and in our study, we have selected the most commonly used and well-documented four methods.”
- Please, add paragraph on the CIMT limitations. This issue was described in depth in J. Clin. Med. 2021, 10(20), 4628; https://doi.org/10.3390/jcm10204628
Response: We have added the following paragraph to stress this point in our limitation on Page 13 Line 6-9.
“First, the use of c-IMT as a primary outcome parameter has inherent limitations. This is due to the presence of both inter- and intra-observer bias when measuring c-IMT. In addition, the comprehensive data analysis and interpretation of results may differ based on the methodology used.“
- Anyway, a routine CIMT measurements is not a standard for the assessment of cardiovascular risk according to current statements of the respective Societies. Please, write why You think that it is worth of measuring.
Response: We have added the following paragraph in the discussion on Page 13 Line 20-26.
“Currently, c-IMT is not a standard method for assessing cardiovascular disease in patients with T2D. However, as previously mentioned, it is a less expensive, non-invasive, and less labor-intensive modality. There have been several studies published in this field, such as the study conducted by Polak et al. In this study, Polak et al. demonstrated that c-IMT could predict future cardiovascular disease in 2,965 subjects of the Framingham Offspring Study after a follow-up period of 7.2 years. Therefore, we believe that c-IMT is still a valuable surrogate measure for evaluating the risk of cardiovascular disease.”
Once again, we would like to thank the reviewer for their time and effort in reviewing our article. Their comments and suggestions have been immensely helpful in improving the quality of our work. We hope that our revised article meets the high standards of your esteemed journal.
Sincerely,
Dong-Feng Yeih
Division of Cardiology, Department of Internal Medicine, Fu Jen Catholic University Hospital; School of Medicine, College of Medicine, Fu Jen Catholic University, New Taipei City 24352, Taiwan, R.O.C.
Tel: 886-2-8512-8888 ext 28751
E-mail: ocean082977@gmail.com
Round 2
Reviewer 1 Report
The authors made convincing arguments for their approach, and addressed the issues that I raised to large extend.
Author Response
Dear Reviewer,
I am writing to express my sincere gratitude for the time, effort, and expertise you dedicated to reviewing our manuscript. Your insightful comments and constructive feedback have been immensely valuable in improving the quality and impact of our research. We greatly appreciate the time and effort you invested in critically evaluating our work. We are grateful for your dedication to maintaining the integrity of the scientific community. Your prompt response allowed us to address the issues you raised efficiently and submit a revised version that incorporates your suggestions, further strengthening the manuscript.
We have invited a Native American doctor, Nsengiyumva Ladislas, to assist us in correcting any grammar errors and enhancing the readability of our manuscript. We are confident that our efforts will enhance the manuscript's clarity and coherence, ultimately improving its chances of being accepted for publication.
Once again, thank you for your invaluable feedback and for sharing your expertise with us. Your commitment to excellence and your dedication to improving the quality of scientific research are truly inspiring.
With deepest appreciation,
Dong-Feng Yeih
Reviewer 3 Report
The paper was significantly improved and I have no further comments.
The paper was significantly improved. It presents important subject on machine learing. I have no concerns regarding English quality.
Author Response

(The authors gave the same response as above.)
